# Robust Reinforcement Learning in Continuous Control Tasks with Uncertainty Set Regularization

**Yuan Zhang**
Neurorobotics Lab
University of Freiburg
yzhang@cs.uni-freiburg.de

**Jianhong Wang**[*]
Center for AI Fundamentals
University of Manchester
jianhong.wang@manchester.ac.uk

**Joschka Boedecker**
Neurorobotics Lab
University of Freiburg
jboedeck@cs.uni-freiburg.de

**Abstract:** Reinforcement learning (RL) is recognized as lacking generalization and robustness under environmental perturbations, which excessively restricts its application for real-world robotics. Prior work claimed that adding regularization to the value function is equivalent to learning a robust policy under uncertain transitions. Although the regularization-robustness transformation is appealing for its simplicity and efficiency, it is still lacking in continuous control tasks. In this paper, we propose a new regularizer named **U**ncertainty **S**et **R**egularizer (USR), to formulate the uncertainty set on the parametric space of a transition function. To deal with unknown uncertainty sets, we further propose a novel adversarial approach to generate them based on the value function. We evaluate USR on the Real-world Reinforcement Learning (RWRL) benchmark and the Unitree A1 Robot, demonstrating improvements in the robust performance of perturbed testing environments and sim-to-real scenarios.

**Keywords:** Reinforcement Learning, Robustness, Continuous Control, Robotics

## 1 Introduction

Reinforcement Learning (RL) is a powerful algorithmic paradigm used to solve sequential decision-making problems and has resulted in great success in various types of environments, e.g., mastering the game of Go [1], playing computer games [2] and operating smart grids [3]. The majority of these successes rely on an implicit assumption that *the testing environment is identical to the training environment*. However, this assumption is too strong for most realistic problems, such as controlling a robot. In more detail, there are several situations where mismatches might appear between training and testing environments in robotics: (1) *Parameter Perturbations* indicates that a large number of environmental parameters, e.g. temperature, friction factor could fluctuate after deployment and thus deviate from the training environment [4]; (2) *System Identification* estimates a transition function from limited experience. This estimation is biased compared with the real-world model [5]; (3) *Sim-to-Real* learns a policy in a simulated environment and performs on real robots for reasons of safety and efficiency [6]. The difference between simulated and realistic environments renders sim-to-real a challenging task.

In this paper, we aim to model the mismatch between training environments and testing environments as a robust RL problem, which regards training environments and testing environments as candidates in an uncertainty set including all possible environments. Robust Markov Decision Pro-

---

[*]Corresponding author.

7th Conference on Robot Learning (CoRL 2023), Atlanta, USA.

cesses (Robust MDPs) [7, 8] is a common theoretical framework to analyze the robustness of RL algorithms. In contrast to the vanilla MDPs with a single transition model $P(s'|s,a)$, Robust MDPs consider an uncertainty set of transition models $\mathbb{P} = \{P\}$ to describe the uncertain environments. This formulation is general enough to cover the various scenarios for the robot learning problems aforementioned.

The aim of Robust RL is to learn a policy under the worst-case scenarios among all transition models $P \in \mathbb{P}$, named *robust policy*. If a transition model $P$ is viewed as an adversarial agent with the uncertainty set $\mathbb{P}$ as its action space, one can reformulate Robust RL as a zero-sum game [9]. In general, solving such a problem is an NP-hard problem [8, 10], however, the employment of Legendre-Fenchel transform can avoid excessive mini-max computations through converting minimization over the transition model to regularization on the value function. Furthermore, it enables more feasibility and tractability to design novel regularizers to cover different types of transition uncertainties. The complexity of these value-function-based regularizers increases with the size of the state space, which leads to a nontrivial extension to continuous control tasks with infinitely large state space. This directly motivates the work of this paper. Due to the page limit, we conclude other related work in Appendix A.

We now summarize the contributions of this paper: (1) the robustness-regularization duality method is extended to continuous control tasks in parametric space; (2) the **U**ncertainty **S**et **R**egularizer (USR) on existing RL frameworks is proposed for learning robust policies; (3) the value function is learnt through an adversarial uncertainty set when the actual uncertainty set is unknown in some scenarios; (4) the USR is evaluated on the Real-world Reinforcement Learning (RWRL) benchmark, showing improvements for robust performances in perturbed testing environments with unknown uncertainty sets; (5) the sim-to-real performance of USR is verified through realistic experiments on the Unitree A1 robot.

## 2 Preliminaries

**Robust MDPs.** The mathematical framework of Robust MDPs [7, 8] extends regular MDPs in order to deal with uncertainty about the transition function. A Robust MDP can be formulated as a 6-tuple $\langle \mathcal{S}, \mathcal{A}, \mathbb{P}, r, \mu_0, \gamma \rangle$, where $\mathcal{S}, \mathcal{A}$ stand for the state and action space respectively, and $r(s,a) : \mathcal{S} \times \mathcal{A} \to \mathbb{R}$ stands for the reward function. Let $\Delta_{\mathcal{S}}$ and $\Delta_{\mathcal{A}}$ be the probability measure on $S$ and $A$ respectively. The initial state is sampled from an initial distribution $\mu_0 \in \Delta_{\mathcal{S}}$, and the future rewards are discounted by the discount factor $\gamma \in [0, 1]$. The most important concept in robust MDPs is the uncertainty set $\mathbb{P} = \{P(s'|s,a)\}$ that controls the variation of transition function $P : \mathcal{S} \times \mathcal{A} \to \Delta_{\mathcal{S}}$, compared with the stationary transition function $P$ in regular MDPs. Let $\Pi = \{\pi(a|s) : \mathcal{S} \to \Delta_{\mathcal{A}}\}$ be the policy space; the objective of Robust RL can then be formulated as a minimax problem such that

$$J^* = \max_{\pi \in \Pi} \min_{P \in \mathbb{P}} \mathbb{E}_{\pi, P} \left[ \sum_{t=0}^{+\infty} \gamma^t r(s_t, a_t) \right]. \tag{1}$$

**Robust Bellman Equation.** While Wiesemann et al. [8] has proved NP-hardness of this minimax problem with an arbitrary uncertainty set, most recent studies [7, 9, 10, 11, 12, 13, 14, 15, 16] approximate it by assuming a rectangular structure on the uncertainty set, i.e., $\mathbb{P} = \times_{(s,a) \in \mathcal{S} \times \mathcal{A}} \mathbb{P}_{sa}$, where $\mathbb{P}_{sa} = \{P(s'|s,a) \quad P_{sa}(s')$ in short$\}$ denotes the local uncertainty of the transition at $(s,a)$. In other words, the variation of transition is independent at every $(s,a)$ pair. Under the assumption of a rectangular uncertainty set, the robust action value $Q^\pi(s,a)$ under policy $\pi$ must satisfy the following robust version of the Bellman equation [17] such that

$$Q^\pi(s,a) = r(s,a) + \min_{P_{sa} \in \mathbb{P}_{sa}} \gamma \sum_{s'} P_{sa}(s')V^\pi(s'), \tag{2}$$

where $V^\pi(s') = \sum_{a'} \pi(a'|s')Q^\pi(s',a')$. Nilim and Ghaoui [11] have shown that a robust Bellman operator admits a unique fixed point of Equation 2, the robust action value $Q^\pi(s,a)$.

**Robustness-Regularization Duality.** Solving the minimization problem in the RHS of Equation 2 can be further simplified by the Legendre-Fenchel transform [18]. For an arbitrary function $f : X \rightarrow \mathbb{R}$, its convex conjugate is $f^*(x^*) := \sup\{\langle x^*, x\rangle - f(x) : x \in X\}$. Define $\delta_{\mathbb{P}_{sa}}(P_{sa}) = 0$ if $P_{sa} \in \mathbb{P}_{sa}$ and $+\infty$ otherwise, Equation 2 can be transformed to its convex conjugate (refer to Derman et al. [10] for detailed derivation) such that

$$Q^\pi(s,a) = r(s,a) + \min_{P_{sa}} \gamma \sum_{s'} P_{sa}(s')V^\pi(s') + \delta_{\mathbb{P}_{sa}}(P_{sa}) = r(s,a) - \delta^*_{\mathbb{P}_{sa}}(-V^\pi(\cdot)). \tag{3}$$

The transformation implies that the robustness condition on transitions can be equivalently expressed as a regularization term on the value function, which is referred to as the robustness-regularization duality. The duality can extensively reduce the cost of solving the minimization problem over infinite transition choices and thus is widely studied in the robust reinforcement learning research community [19, 20, 21].

As a special case, Derman et al. [10] considered a $L_2$ norm uncertainty set on transitions, i.e., $\mathbb{P}_{sa} = \{\bar{P}_{sa} + \alpha\tilde{P}_{sa} : \|\tilde{P}_{sa}\|_2 \leq 1\}$, where $\bar{P}_{sa}$ is usually called the nominal transition model. It can represent the prior knowledge of a transition model or a numerical value of the training environment. The uncertainty set implies that the transition model is allowed to fluctuate around the nominal model with some degree $\alpha$. Therefore, the corresponding Bellman equation in Equation 3 becomes $Q^\pi(s,a) = r(s,a) + \gamma \sum_{s'} \bar{P}_{sa}(s')V^\pi(s') - \alpha\|V^\pi(\cdot)\|_2$. Similarly, the $L_1$ norm has also been used as uncertainty set on transitions [14], i.e., $\mathbb{P}_{sa} = \{\bar{P}_{sa} + \alpha\tilde{P}_{sa} : \|\tilde{P}_{sa}\|_1 \leq 1\}$, and the Bellman equation becomes the form such that $Q^\pi(s,a) = r(s,a) + \gamma \sum_{s'} \bar{P}_{sa}(s')V^\pi(s') - \alpha \max_{s'} |V^\pi(s')|$. This robustness-regularization duality works well in the finite state space but the extension to the infinite state space is still a question. We claim that such an extension is non-trivial, since both regularizers $\|V^\pi(\cdot)\|_2$ and $\max_{s'} |V^\pi(s')|$ need to be calculated on the infinite-dimensional vector $V^\pi(\cdot)$. In this work, we extend this concept to the continuous state space which is a critical characteristic in robotics.

## 3 Uncertainty Set Regularized Robust Reinforcement Learning

Having introduced the robustness-regularization duality and the difficulties regarding its extension to the continuous state space in Section 2, here, we will first present a novel extension to the continuous state space with the uncertainty set defined on the parametric space of a transition function. We will then utilize this extension to derive a robust policy evaluation method that can be directly plugged into existing RL algorithms. Furthermore, to deal with the unknown uncertainty set, we propose the adversarial uncertainty set and visualize it in a simple *moving-to-target* task.

### 3.1 Uncertainty Set Regularized Robust Bellman Equation (USR-RBE)

For environments with continuous state space, the transition model $P(s'|s,a)$ is usually represented as a parametric function $P(s'|s,a;w)$, where $w$ denotes the parameters of the transition function. Instead of defining the uncertainty set on the distribution space, we directly impose a perturbation on $w$ within a set $\Omega_w$. Consequently, the robust objective function (Equation 1) becomes $J^* = \max_{\pi \in \Pi} \min_{w \in \Omega_w} \mathbb{E}_{\pi, P(s'|s,a;w)} \left[\sum_{t=0}^{+\infty} \gamma^t r(s_t, a_t)\right]$. We further assume that the parameter $w$ fluctuates around a nominal parameter $\bar{w}$, such that $w = \bar{w} + \tilde{w}$, with $\bar{w}$ being a fixed parameter and $\tilde{w} \in \Omega_{\tilde{w}} = \{w - \bar{w}|w \in \Omega_w\}$ being the perturbation part. Inspired by Equation 3, we can derive a robust Bellman equation on the parametric space for continuous control problems as shown in Proposition 3.1.

**Proposition 3.1 (Uncertainty Set Regularized Robust Bellman Equation)** *Suppose the uncertainty set of $w$ is $\Omega_w$ (i.e., the uncertainty set of $\tilde{w} = w - \bar{w}$ is $\Omega_{\tilde{w}}$), the robust Bellman equation on the parametric space can be represented as follows:*

$$Q^\pi(s,a) = r(s,a) + \gamma \int_{s'} P(s'|s,a;\bar{w})V^\pi(s')ds' - \gamma \int_{s'} \delta^*_{\Omega_{\tilde{w}}} \left[-\nabla_w P(s'|s,a;\bar{w})V^\pi(s')\right] ds', \tag{4}$$

where $\delta_{\Omega_w}(w)$ is the indicator function that equals $0$ if $w \in \Omega_w$ and $+\infty$ otherwise, and $\delta^*_{\Omega_w}(w')$ is the convex dual function of $\delta_{\Omega_w}(w)$.

The proof is presented in Appendix B.1. Intuitively, Proposition 3.1 shows that ensuring robustness on parameter $w$ can be transformed into regularization on action value $Q^\pi(s, a)$ that relates to the product of the state value function $V^\pi(s')$ and the derivative of the transition model $\nabla_w P(s'|s, a; \bar{w})$. Taking the $L_2$ uncertainty set (also used in Derman et al. [10]) as a special case, i.e., $\Omega_w = \{\bar{w} + \alpha\tilde{w} : \|\tilde{w}\|_2 \leq 1\}$, where $\bar{w}$ stands for the parameter of the nominal transition model $P(s'|s, a'; \bar{w})$, the robust Bellman equation in Proposition 3.1 becomes

$$Q^\pi(s, a) = r(s, a) + \gamma \int_{s'} P(s'|s, a; \bar{w})V^\pi(s')ds' - \alpha \int_{s'} \|\nabla_w P(s'|s, a; \bar{w})V^\pi(s')\|_2 ds'. \quad (5)$$

### 3.2 Uncertainty Set Regularized Robust Reinforcement Learning (USR-RRL)

To derive a practical robust RL algorithm using the proposed USR-RBE, we follow the policy iteration framework [22] commonly used in RL research. Regarding the policy evaluation procedure, Theorem 3.2 proposes an operator on action value $Q^\pi$ and ensures its convergence to a unique fixed point by recursively running this operator. The proof can be found in Appendix B.2. Intuitively, this theorem indicates that one can acquire a robust action value given a certain policy and uncertainty set if the discount factor $\gamma$ and the coefficient in the uncertainty set $\alpha$ are properly set, which is satisfied in the practical implementations. As for the policy improvement procedure, all existing techniques (e.g. policy gradient methods) can be adopted to optimize the policy. By iterating the policy evaluation and improvement cycle, the policy will eventually converge to an equilibrium trading off optimality and robustness.

**Theorem 3.2 (Convergence of Robust Policy Evaluation)** *For any policy $\pi \in \Delta_\mathcal{A}$, the following operator $T$ can reach a unique fixed point as the robust action value $Q^\pi(s, a)$ if $0 \leq \gamma + \alpha \max_{s,a} \left| \int_{s'} \|\nabla_w P(s'|s, a; \bar{w})\|_2 ds' \right| \leq 1$.*

$$TQ^\pi(s, a) = r(s, a) + \gamma \int_{s'} P(s'|s, a; \bar{w})V^\pi(s')ds' - \alpha \int_{s'} \|\nabla_w P(s'|s, a; \bar{w})V^\pi(s')\|_2 ds',$$

*where $V^\pi(s') = \int_{s'} \pi(a'|s')Q(s', a')da'$.*

A practical concern on this algorithm is that calculating USR-RBE requires knowledge of the transition model $P(s'|s, a; \bar{w})$. This naturally applies to model-based RL, as model-based RL learns a point estimate of the transition model $P(s'|s, a; \bar{w})$ by maximum likelihood approaches [23]. For model-free RL, we choose to construct a local Gaussian model with mean as parameters inspired by previous work [24]. Specifically, suppose that one can access a triple (state $s$, action $a$ and next state $x$) from the experience replay buffer, then a local transition model $P(s'|s, a; \bar{w})$ can be modelled as a Gaussian distribution with mean $x$ and covariance $\Sigma$ (with $\Sigma$ being a hyperparameter), i.e., the nominal parameter $\bar{w}$ consists of $(x, \Sigma)$. With this local transition model, we now have the full knowledge of $P(s'|s, a; \bar{w})$ and $\nabla_w P(s'|s, a; \bar{w})$, which allows us to calculate the RHS of Equation 5. To further approximate the integral calculation in Equation 5, we sample $M$ points $\{s'_1, s'_2, ..., s'_M\}$ from the local transition model $P(s'|s, a; \bar{w})$ and use them to approximate the target action value by $Q^\pi(s, a) \approx r(s, a) + \gamma \sum_{i=1}^{M} \left[ V^\pi(s'_i) - \alpha\|\nabla_w P(s'_i|s, a; \bar{w})V^\pi(s_i)\|_2 / P(s'_i|s, a; \bar{w}) \right]$, where $\bar{w} = (x, \Sigma)$. With this approximation, the Bellman operator is guaranteed to converge to the robust value, and policy improvement is applied to robustly optimize the policy. We explain how to incorporate USR-RBE into a classic RL algorithm called Soft Actor Critic (SAC) [25] in Appendix B.3.

### 3.3 Adversarial Uncertainty Set

The proposed method in Section 3.2 relies on the prior knowledge of the uncertainty set of the parametric space. The $L_p$ norm uncertainty set is most widely used in the Robust RL and robust optimal

control literature. However, such a fixed uncertainty set may not sufficiently adapt to various perturbation types. The $L_p$ norm uncertainty set with its larger region can result in an over-conservative policy, while the one with a smaller region may lead to a risky policy. In this section, we learn an adversarial uncertainty set through the agent's policy and value function to avoid the above issues.

**Generating the Adversarial Uncertainty Set.** The basic idea of the adversarial uncertainty set is to provide an appropriate uncertainty range to parameters that are more sensitive to the value function, which is naturally measured by the derivative. The agent learning based on such an adversarial uncertainty set is easier to adapt to the various perturbation types of parameters. We generate the adver-

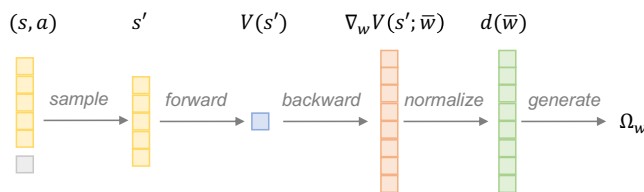

Figure 1: Illustration of the procedure for generating an adversarial uncertainty set.

sarial uncertainty set in a 5-step procedure as illustrated in Figure 1, (1) *sample* next state $s'$ according to the distribution $P(\cdot|s, a; \bar{w})$, given the current state $s$ and action $a$; (2) *forward* pass by calculating the state value $V(s')$ at next state $s'$; (3) *backward* pass by using the reparameterization trick [26] to compute the derivative $g(\bar{w}) = \nabla_w V(s'; \bar{w})$; (4) *normalize* the derivative by $d(\bar{w}) = g(\bar{w})/[\sum_i^W g(\bar{w})_i^2]^{0.5}$; (5) *generate* the adversarial uncertainty set $\Omega_w = \{\bar{w} + \alpha\tilde{w} : \|\tilde{w}/d(\bar{w})\|_2 \leq 1\}$. The pseudo-code to generate the adversarial uncertainty set is explained in Appendix B.4 Algorithm 2.

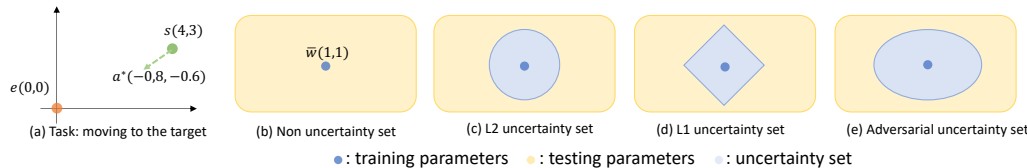

(a) Task: moving to the target    (b) Non uncertainty set    (c) L2 uncertainty set    (d) L1 uncertainty set    (e) Adversarial uncertainty set

● : training parameters    ● : testing parameters    ● : uncertainty set

Figure 2: Illustration of different types of uncertainty sets to investigate their characteristics.

**Characteristics of the Adversarial Uncertainty Set.** To further investigate the characteristics of the adversarial uncertainty set, we visualize it in a simple *moving-to-target* task: controlling a particle to move towards a target point $e$ (Figure 2.a). The two-dimensional state $s$ informs the position of the agent, and the two-dimensional action $a = (a_1, a_2)$ ($\|a\|_2 = 1$ is normalized) controls the force in two directions. The environmental parameter $w = (w_1, w_2)$ represents the contact friction in two directions respectively. The transition function is expressed as $s' \sim \mathcal{N}(s + (a_1 w_1, a_2 w_2), \Sigma)$, and the reward is defined by the difference of the distances to the target point at two successive steps minus a stage cost: $r(s, a, s') = d(s, e) - d(s', e) - 2$. The nominal value $\bar{w} = (1, 1)$ (Figure 2.b) indicates the equal friction factor in two directions for the training environment. It is easy to conjecture that the optimal action is pointing towards the target point, and the optimal value function is $V^*(s) = -d(s, e)$. We visualize $L_2$, $L_1$ and adversarial uncertainty set of the contact friction $w$ in Figure 2.(c,d,e) respectively, at a specific state $s = (4, 3)$ and the optimal action $a^* = (-0.8, -0.6)$. The uncertainty sets $L_2$ and $L_1$ satisfy $(w_1^2 + w_2^2)^{0.5} \leq 1$ and $|w_1| + |w_2| \leq 1$ respectively. Adversarial uncertainty set is calculated by following the generation procedure described in Section 3.3, where the normalized derivative $d(\bar{w})$ is $[0.8, 0.6]^T$ and adversarial uncertainty set is $(w_1^2/0.64 + w_2^2/0.36)^{0.5} \leq 1$, as an ellipse in Figure 2.e. Compared with the $L_2$ uncertainty set, the adversarial uncertainty set extends the perturbation range of the horizontal dimensional parameter since it could be more sensitive to the final return. As a result, the agent learning to resist such an uncertainty set is expected to perform well on unseen perturbation types, which will be verified in more realistic experiments in the next section.

# 4 Experiments

In this section, we provide experimental results on the Real-world Reinforcement Learning (RWRL) benchmark [27], to validate the effectiveness of the proposed regularizing USR for resisting perturbations in the environment. Besides, we apply USR on a sim-to-real task to show its potential on real-world robots.

## 4.1 Experimental Setups

**Task Description.** RWRL, whose back-end is the Mujoco environment [28], is a continuous control benchmark consisting of real-world challenges for RL algorithms. Using this benchmark, we will evaluate the proposed algorithm regarding the robustness of the learned policy in physical environments with perturbations of parameters of the state equations (dynamics). In more detail, we first train a policy through interaction with the nominal environments (i.e., the environments without any perturbations), and then test the policy in the environments with perturbations within a range over relevant physical parameters. The setup is visualized as a process map in Appendix C.1. In this paper, we conduct experiments on six tasks: *cartpole_balance*, *cartpole_swingup*, *walker_stand*, *walker_walk*, *quadruped_walk*, *quadruped_run*, with growing complexity in state and action space. More details about the specifications of tasks are shown in Appendix C.2. The perturbed variables and their value ranges can be found in Table 2.

**Evaluation metric.** A severe problem for Robust RL research is the lack of a standard metric to evaluate policy robustness. To resolve this obstacle, we define a new robustness evaluation metric which we call *Robust-AUC* to calculate the area under the curve of the return with respect to the perturbed physical variables, in analogy to the definition of regular AUC [29]. More specifically, a trained policy $\pi$ is evaluated in an environment with perturbed variable $P$ whose values $v$ change in the range $[v_{min}, v_{max}]$ and achieves different returns $r$. Then, these two sets of data are employed to draw a parameter-return curve $C(v, r)$ to describe the relationship between returns $r$ and perturbed values $v$. We define the relative area under this curve as *Robust-AUC* such that $Robust\text{-}AUC = \frac{Area(C(v,r))}{v_{max} - v_{min}}, v \in [v_{min}, v_{max}]$. Compared to the vanilla AUC, *Robust-AUC* describes the correlation between returns and the perturbed physical variables, which can sensitively reflect the response of a learning procedure (to yield a policy) to unseen perturbations, i.e., the robustness. We further explain the practical implementations to calculate *Robust-AUC* of the experiments in Appendix C.3.

**Baselines and Implementation of Proposed Methods.** We first compare USR with a standard version of Soft Actor Critic (SAC) [25], which stands for the category of algorithms without regularizers (*None-Reg*). Another category of baselines is to directly impose $L_p$ regularization on the parameters of the value function (*L1-Reg*, *L2-Reg*) [30], which is a common way to improve the generalization of function approximation but without consideration of the environmental perturbations; For a fixed uncertainty set as introduced in Section 3.2, we implement two types of uncertainty sets on transitions, *L2-USR* and *L1-USR*, which can be viewed as an extension of Derman et al. [10] and Wang and Zou [14] for continuous control tasks respectively; finally, we also evaluate the adversarial uncertainty set (Section 3.3), denoted as *Adv-USR*. We conclude all model structures and hyperparameters in Appendix C.4 - C.5. The code of experiments is available on github.com/mikezhang95/rrl_usr.

## 4.2 Main Results

We show the *Robust-AUC* and its significance value of *cartpole_swingup*, *walker_stand*, *quadruped_walk* in Table 1. Due to page limits, the results of other tasks are presented in Appendix D.1. In addition to calculating *Robust-AUC* under different perturbations, we also rank all algorithms and report the average rank as an overall robustness performance of each task. Notably, *L1-Reg* and *L2-Reg* do not improve on the robustness, and even impair the performance in comparison with the None-Regularized agent on simple domains (*cartpole* and *walker*). In contrast, we

Table 1: *Robust-AUC* of all algorithms and their uncertainties on RWRL benchmark.

| Task Name | Variables | Algorithms | | | | | |
|---|---|---|---|---|---|---|---|
| | | None-Reg | L1-Reg | L2-Reg | L1-USR | L2-USR | Adv-USR |
| *cartpole_swingup* | pole_length | 393.41 (21.95) | 319.73 (23.84) | 368.16 (6.33) | **444.93 (12.77)** | 424.48 (2.27) | 430.91 (2.35) |
| | pole_mass | 155.25 (4.79) | 96.85 (4.32) | 131.35 (12.42) | 175.28 (3.33 ) | 159.61 (2.41) | **193.13 (2.27)** |
| | joint_damping | 137.20 (4.55) | 140.16 (0.58) | 165.01 (0.16) | 164.21 (0.48) | 169.88 (2.68) | **170.39 (0.76)** |
| | slider_damping | 783.76 (9.14) | 775.73 (16.50) | 797.59 (5.52) | 793.55 (5.02) | 781.02 (5.80) | **819.32 (3.00)** |
| | average rank | 4.5 | 5.75 | 3.75 | 2.5 | 3.25 | **1.25** |
| *walker_stand* | thigh_length | 487.02 (40.50) | 461.95 (50.03) | 497.38 (43.98) | 488.71 (42.42) | **511.16 (49.84)** | 505.88 (50.22) |
| | torso_length | 614.06 (41.10) | 586.16 (68.84) | 586.20 (44.31) | 598.02 (36.17) | 610.93 (38.87) | **623.56 (46.47)** |
| | joint_damping | **607.24 (115.28)** | 387.89 (109.53) | 443.82 (63.52) | 389.77 (76.96) | 527.87 (116.75) | 514.77 (126.00) |
| | contact_friction | 946.74 (22.20) | **947.24 (29.16)** | 941.92 (22.21) | 943.11 (21.97) | 940.73 (20.89) | 945.69 (16.02) |
| | average rank | 2.50 | 4.75 | 4.25 | 4.25 | 3.00 | **2.25** |
| *quadruped_walk* | shin_length | 492.55 (124.44) | 406.77 (90.41) | 503.13 (106.50) | 540.39 (126.22) | 564.60 (135.49) | **571.85 (64.99)** |
| | torso_density | 471.45 (99.14) | 600.86 (45.21) | 526.22 (79.11) | 442.05 (70.73) | 472.80 (83.22) | **602.09 (55.36)** |
| | joint_damping | 675.95 (67.23) | 711.54 (91.84) | **794.56 (65.64)** | 762.50 (79.76) | 658.17 (112.67) | 785.11 (40.79) |
| | contact_friction | 683.80 (135.80) | 906.92 (100.19) | 770.44 (158.42) | 777.40 (106.04) | 767.80 (109.00) | **969.73 (21.24)** |
| | average rank | 5.25 | 3.50 | 3.00 | 3.75 | 4.25 | **1.25** |

observe that both *L2-USR* and *L1-USR* can outperform the default version under certain perturbations (e.g. *L1-USR* in *cartpole_swingup* for pole_length, *L2-USR* in *walker_stand* for thigh_length); they are, however, not effective for all scenarios. We argue that the underlying reason could be that the fixed shape of the uncertainty set cannot adapt to all perturbed cases. This is supported by the fact that *Adv-USR* achieves the best average rank among all perturbed scenarios, showing the best zero-shot generalization performance in continuous control tasks. For complex tasks like *quadruped_run*, it is surprising that *L1-Reg* can achieve competitive results compared with *Adv-USR* but with slightly larger uncertainty on the *Robust-AUC*, probably because the sparsity by L1 regularization can reduce redundant features. We also compare the computational cost of all algorithms both empirically and theoretically in Appendix D.2. It is concluded that *Adv-USR* can improve the robustness in most cases without increasing too much computational burden. More limitations on the computational time are discussed in Section 6. We also carry out two additional testing scenarios to imitate the perturbation in real: all parameters deviate from the nominal values simultaneously (Appendix D.3) and the perturbed value follows a random walk during the testing episode (Appendix D.4). *Adv-USR* consistently performs best and is well-adapted to different perturbations. The additional analysis on the training process of *Adv-USR* can be found in Appendix D.5.

## 4.3  Study on Sim-to-real Robotic Task

In this section, we further investigate the robustness of *Adv-USR* on a sim-to-real robotic task. Sim-to-real is a commonly adopted setup to apply RL algorithms on real robots, where the agent is first trained in simulation and then transferred to real robots. Unlike the environmental setup in Section 4.1 with additional perturbations during the testing phases, sim-to-real inherently possesses a mismatch between training and testing environ-

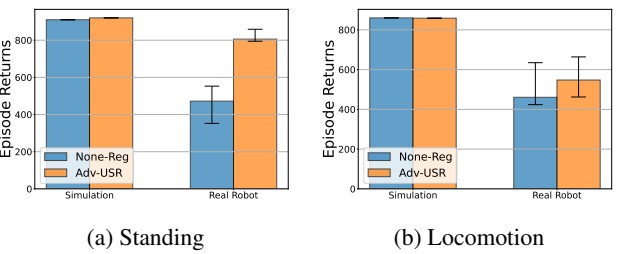

(a) Standing  (b) Locomotion

Figure 3: Episode returns of all algorithms on sim-to-real task.

ments potentially due to: (1) the simulator possesses a simplified dynamics model and suffers from accumulated error [31] and (2) there are significant differences between simulators and real hardware in robot's parameters, such as a quadruped example in Table 5. As a result, this setup is an ideal testbed and practical application of the proposed robust RL algorithm.

Specifically, we use the Unitree A1 robot [32] and the Bullet simulator [33] as the platform for sim-to-real transfer. The agents learn standing and locomotion in simulations and directly perform on real robots without adaption [34]. Since other baselines cannot generalize well even in pure simu-

lated RWRL environments, we only compare SAC agents with and without *Adv-USR* method. Most previous works [35, 36] utilize domain randomization (DR) techniques [37] to deal with sim-to-real mismatches. DR requires training on multiple randomly initialized simulated instances with diverse environmental parameters, expecting the policy to be generalized in the testing environment (real robot). In contrast, *Adv-USR* only requires training on single nominal parameters, which tremendously improves the efficiency and feasibility. Detailed setup of the sim-to-real task is described in Appendix C.6. We also additionally compare DR theoretically and empirically in Appendix E.

We run 50 testing trials per baseline and report the episodic returns in Figure 3. Both agents succeed in learning a nearly optimal policy in simulation for both tasks. For the standing task, the agent with *Adv-USR* maintains its performance on real robots while the other agent fails to achieve the standing position. For the more complex locomotion task, we notice that the observation is hugely different from simulation and real robots as the position and velocity estimators are noisy and delayed and parameters in Table 5 are varied. That's why *None-Reg* hardly moves any legs on real robots as in Figure 4b. But *Adv-USR* reveals a certain level of robustness by iterative bending and moving all legs (Figure 4a) and surpassing velocity (Figure 4c). However, the performance still deteriorates compared with the simulation. To alleviate the extreme sim-to-real difference, combining *Adv-USR* and other sim-to-real techniques could be a more powerful strategy, which would be verified in further work.

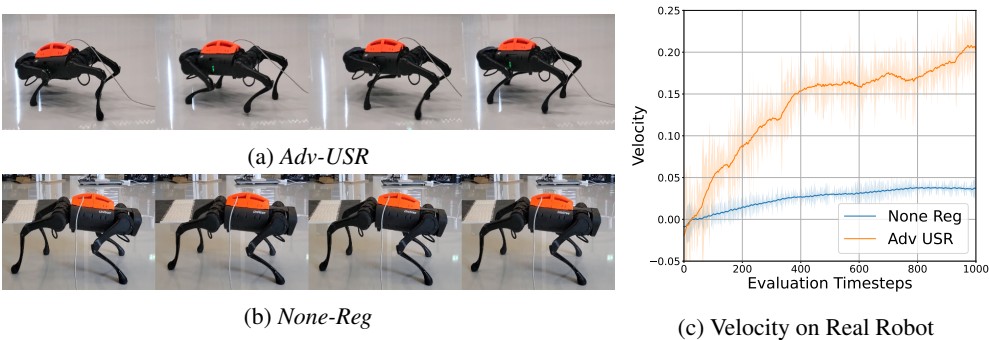

(a) *Adv-USR*

(b) *None-Reg*

(c) Velocity on Real Robot

Figure 4: Agent behaviours of all algorithms on locomotion task.

# 5   Conclusion

In this paper, we adopt the robustness-regularization duality method to design new regularizers for continuous control problems to improve the robustness and generalization of RL algorithms. Furthermore, to deal with unknown uncertainty sets, we design an adversarial uncertainty set depending on the learned action state value function and implement it as a new regularizer. The proposed method shows great promise regarding generalization and robustness under environmental perturbations in both simulated and realistic robotic tasks. Noticeably, it does not require training in multiple diverse environments or fine-tuning in testing environments, which makes it an efficient and valuable add-on to RL for robot learning.

# 6   Limitations

The limitations of this work are discussed as follows. First, although the computational cost of *Adv-USR* is acceptable [24] for the local Gaussian model owing to low-dimensional proprioceptive observations (e.g. position, velocity) in the experiments, it is a critical factor when *Adv-USR* is applied to more complex dynamics with millions of parameters (i.e. common in recent offline and model-based RL research [38]). Therefore, methods to automatically detect critical variables may be required in future work. On the other hand, it is probable to extend *Adv-USR* with domain randomization to tackle more sophistical robotic tasks in the future.

**Acknowledgments**

This research receives funding from the European Union's Horizon 2020 research and innovation program under the Marie Skłodowska-Curie grant agreement No. 953348 (ELO-X). Jianhong Wang is fully supported by UKRI Turing AI World-Leading Researcher Fellowship, EP/W002973/1. The authors thank Jasper Hoffman, Baohe Zhang for the inspiring discussions.

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

# A    Related Work

Robust Reinforcement Learning (Robust RL) has recently become a popular topic [7, 27, 39, 40], due to its effectiveness in tackling perturbations. Besides the transition perturbation in this paper, there are other branches relating to action, state and reward. We will briefly discuss them in the following paragraphs. Additionally, we will discuss the relation of Robust RL, sim-to-real, Bayesian RL and Adaptive RL approaches, which are also important topics in robot learning

**Action Perturbation.** Early works in Robust RL concentrated on action space perturbations. Pinto et al. [41] first proposed an adversarial agent perturbing the action of the principle agent, training both alternately in a mini-max style. Tessler et al. [42] later performed action perturbations with probability $\alpha$ to simulate abrupt interruptions in the real world. Afterwards, Kamalaruban et al. [43] analyzed this mini-max problem from a game-theoretic perspective and claimed that an adversary with mixed strategy converges to a mixed Nash Equilibrium. Similarly, Vinitsky et al. [44] involved multiple adversarial agents to augment the robustness, which can also be explained in the view of a mixed strategy.

**State Perturbation.** State perturbation can lead to the change of state from $s$ to $s_p$, and thus might worsen an agent's policy $\pi(a|s)$ [45]. Zhang et al. [46], Oikarinen et al. [47] both assume an $L_p$-norm uncertainty set on the state space (inspired by the idea of adversarial attacks widely used in computer vision [48]) and propose an auxiliary loss to encourage learning to resist such attacks. It is worth noting that state perturbation is a special case of transition perturbation, which can be covered by the framework proposed in this paper, as further explained in Appendix F.

**Reward Perturbation.** The robustness-regularization duality has been widely studied, especially when considering reward perturbations [19, 20, 21]. One reason is that the policy regularizer is closely related to a perturbation on the reward function without the need for a rectangular uncertainty assumption. However, it restricts the scope of these works as reward perturbation, since it can be shown to be a particular case of transition perturbation by augmenting the reward value in the state [20]. Besides, the majority of works focus on the analysis of regularization to robustness, which can only analyze the effect of existing regularizers instead of deriving novel regularizers for robustness as in the work we present here.

**Sim-to-real.** Sim-to-real is a key research topic in robot learning. Compared to the Robust RL problem, it aims to learn a robust policy from simulations for generalization in real-world environments. Domain randomization is a common approach to ease this mismatch in sim-to-real problems [6, 37]. However, Mankowitz et al. [13] has demonstrated that it actually optimizes the average case of the environment rather than the worst-case scenario (as seen in our research), which fails to perform robustly during testing. More recent active domain randomization methods [49] resolve this flaw by automatically selecting difficult environments during the training process. The idea of learning an adversarial uncertainty set considered in this paper can be seen as a strategy to actively search for more valuable environments for training.

**Bayesian RL.** One commonality between Bayesian RL and Robust RL is that they both store uncertainties over the environmental parameter (posterior distribution $q(w)$ in Bayesian RL and uncertainty set $\Omega_w$ in Robust RL). Uncertainties learned in Bayesian RL can benefit Robust RL in two ways: (1) Robust RL can define an uncertainty set $\Omega_w = \{w : q(w) > \alpha\}$ to learn a robust policy that can tolerate model errors, which is attractive for offline RL and model-based RL; (2) A soft robust objective with respect to the distribution $q(w)$ can ease the conservative behaviours caused by the worst-case scenario [16].

**Adaptive RL.** Adaptive RL (often referred as Meta RL [50]) is another popular technique to deal with the perturbations in environments parallel to Robust RL introduced in this paper. The main difference between Robust RL and Adaptive RL is whether policy parameters are allowed to change when environmental parameters vary. Robust RL is a zero-shot learning technique aiming to learn one single robust policy that can be applied to various perturbed environments. Adaptive RL is a few-shot learning technique aiming to quickly change policy to adapt to the changing environments.

These two techniques can be combined to increase the robustness in real-world robots. One can first use Robust RL to learn a base policy as the warm start and fine-tune the policy on certain perturbed environments with Adaptive RL.

# B  Extra Algorithm Details

## B.1  Proof of Uncertainty Set Regularized Robust Bellman Equation

The proof is as follows:

$$
\begin{aligned}
Q^\pi(s,a) &= r(s,a) + \gamma \min_{w \in \Omega_w} \int_{s'} P(s'|s,a;w)V^\pi(s')ds' \\
&= r(s,a) + \gamma \int_{s'} P(s'|s,a;\bar{w})V^\pi(s')ds' + \gamma \min_{w \in \Omega_w} \int_{s'} (w-\bar{w})^T \nabla_w P(s'|s,a;\bar{w})V^\pi(s')ds' \\
&= r(s,a) + \gamma \int_{s'} P(s'|s,a;\bar{w})V^\pi(s')ds' - \gamma \left[ \max_{\tilde{w}} \int_{s'} -\tilde{w}^T \nabla_w P(s'|s,a;\bar{w})V^\pi(s')ds' - \delta_{\Omega_{\tilde{w}}}(\tilde{w}) \right] \\
&= r(s,a) + \gamma \int_{s'} P(s'|s,a;\bar{w})V^\pi(s')ds' - \gamma \int_{s'} \delta^*_{\Omega_{\tilde{w}}} \left[ -\nabla_w P(s'|s,a;\bar{w})V^\pi(s') \right] ds'
\end{aligned}
$$

$$\tag{6}$$

The second line utilizes the first-order Taylor Expansion at $\bar{w}$. The third line reformulates the minimization on $w$ to maximization on $\tilde{w}$ and adds an indicator function as a hard constraint on $\tilde{w}$. The last line directly follows the definition of convex conjugate function.

## B.2  Convergence of Robust Policy Evaluation

We now prove that the Bellman operator with an extra regularizer term as the *policy evaluation* stage can converge to the robust action value function given some specific conditions.

Since $V^\pi(s) = \int_a \pi(a|s)Q^\pi(s,a)da$, we define the equivalent operator with respect to $Q^\pi$ to the one proposed in this paper as $T$ such that

$$
\begin{aligned}
TQ^\pi(s,a) &= r(s,a) + \gamma \int_{s'} P(s'|s,a;\bar{w}) \int_{a'} Q^\pi(s',a')da'ds' - \alpha \int_{s'} \|\nabla_w P(s'|s,a;\bar{w}) \int_{a'} Q^\pi(s',a')da'\|_2 ds' \\
&= \underbrace{r(s,a) + \gamma \int_{s'} P(s'|s,a;\bar{w})V^\pi(s')ds' - \alpha \int_{s'} \|\nabla_w P(s'|s,a;\bar{w})V^\pi(s')\|_2 ds'}_{\text{The operator proposed in this paper.}} .
\end{aligned}
$$

$$\tag{7}$$

To ease the derivation, we will not expand the term $V^\pi$ as the form of $Q^\pi$ at the beginning in the following procedures.

$$\|TQ_1^\pi - TQ_2^\pi\|_\infty = \max_{s,a} \left| r(s,a) + \gamma \int_{s'} P(s'|s,a;\bar{w})V_1^\pi(s')ds' - \alpha \int_{s'} \|\nabla_w P(s'|s,a;\bar{w})V_1^\pi(s')\|_2 ds' \right.$$

$$\left. - r(s,a) - \gamma \int_{s'} P(s'|s,a;\bar{w})V_2^\pi(s')ds' + \alpha \int_{s'} \|\nabla_w P(s'|s,a;\bar{w})V_2^\pi(s')\|_2 ds' \right|$$

$$= \max_{s,a} \left| \gamma \int_{s'} P(s'|s,a;\bar{w})\Big[V_1^\pi(s') - V_2^\pi(s')\Big]ds' + \right.$$

$$\left. \alpha \int_{s'} \Big[\|\nabla_w P(s'|s,a;\bar{w})V_2^\pi(s')\|_2 - \|\nabla_w P(s'|s,a;\bar{w})V_1^\pi(s')\|_2\Big]ds' \right|$$

$$\leq \gamma \underbrace{\max_{s,a} \left| \int_{s'} P(s'|s,a;\bar{w})\Big[V_1^\pi(s') - V_2^\pi(s')\Big]ds' \right|}_{\text{Partition 1.}} +$$

$$\underbrace{\alpha \max_{s,a} \left| \int_{s'} \Big[\|\nabla_w P(s'|s,a;\bar{w})V_2^\pi(s')\|_2 - \|\nabla_w P(s'|s,a;\bar{w})V_1^\pi(s')\|_2\Big]ds' \right|}_{\text{Partition 2.}}$$

Since Partition 1 is a conventional Bellman operator, following the contraction mapping property we can directly write the inequality such that

$$\gamma \max_{s,a} \left| \int_{s'} P(s'|s,a;\bar{w})\Big[V_1^\pi(s') - V_2^\pi(s')\Big]ds' \right| \leq \gamma\|V_1^\pi(s') - V_2^\pi(s')\|_\infty. \qquad (8)$$

Next, we will process Partition 2. The details are as follows:

$$\alpha \max_{s,a} \left| \int_{s'} \Big[\|\nabla_w P(s'|s,a;\bar{w})V_2^\pi(s')\|_2 - \|\nabla_w P(s'|s,a;\bar{w})V_1^\pi(s')\|_2\Big]ds' \right|$$

$$\leq \alpha \max_{s,a} \left| \int_{s'} \left\|\nabla_w P(s'|s,a;\bar{w})\Big[V_2^\pi(s') - V_1^\pi(s')\Big]\right\|_2 ds' \right|$$

$$\leq \alpha \max_{s,a} \left| \int_{s'} \|\nabla_w P(s'|s,a;\bar{w})\|_2 |V_2^\pi(s') - V_1^\pi(s')|ds' \right|$$

$$\leq \alpha \max_{s,a} \left| \max_s |V_2^\pi(s') - V_1^\pi(s')| \int_{s'} \|\nabla_w P(s'|s,a;\bar{w})\|_2 ds' \right|$$

$$\leq \alpha \underbrace{\max_{s,a} \left| \int_{s'} \|\nabla_w P(s'|s,a;\bar{w})\|_2 ds' \right|}_{:=\delta} \max_{s'} |V_1^\pi(s') - V_2^\pi(s')|$$

$$= \delta\|V_1^\pi - V_2^\pi\|_\infty. \qquad (9)$$

Combining the results of Eq.8 and Eq.9 and expanding the term $V^\pi$, we can directly get that

$$\|TQ_1^\pi - TQ_2^\pi\|_\infty \leq (\gamma + \delta)\|V_1^\pi - V_2^\pi\|_\infty$$
$$\leq (\gamma + \delta)\max_{s'} |V_1^\pi(s') - V_2^\pi(s')|$$
$$\leq (\gamma + \delta)\max_{s',a'} |Q_1^\pi(s',a') - Q_2^\pi(s',a')|$$
$$\leq (\gamma + \delta)\|Q_1^\pi - Q_2^\pi\|_\infty$$

To enable $T$ to be a contraction mapping in order to converge to the exact robust Q-values, we have to let $0 \leq \gamma + \delta \leq 1$. In more details, the norm of the gradient of transition function with respect to the uncertainty set (i.e., $\|\nabla_w P(s'|s,a;\bar{w})\|_2$ inside $\delta$) is critical to the convergence of robust Q-values under some robust policy. Given the above conditions, the robust value function will finally converge.

### B.3 Algorithm on Incorporating Uncertainty Set Regularizer into Soft Actor Critick

The proposed Uncertainty Set Regularizer method is flexible to be plugged into any existing RL frameworks as introduced in Section 3.2. Here, we include a specific implementation on Soft Actor Critic algorithm, see Algorithm 1.

---

**Algorithm 1** Uncertainty Set Regularized Robust Soft Actor Critic

---

1: Input: initial State $s$, action value $Q(s, a; \theta)$'s parameters $\theta_1$, $\theta_2$, policy $\pi(a|s; \phi)$'s parameters $\phi$, replay buffer $\mathcal{D} = \emptyset$, transition nominal parameters $\bar{w}$, value target update rate $\rho$
2: Set target value parameters $\theta_{tar,1} \leftarrow \theta_1$, $\theta_{tar,2} \leftarrow \theta_2$
3: **repeat**
4:     Execute $a \sim \pi(a|s; \phi)$ in the environment
5:     Observe reward $r$ and next State $s'$
6:     $\mathcal{D} \leftarrow \mathcal{D} \cup \{s, a, r, s'\}$
7:     $s \leftarrow s'$
8:     **for** each gradient step **do**
9:         Randomly sample a batch transitions, $\mathcal{B} = \{s, a, r, s'\}$ from $\mathcal{D}$
10:        Construct adversarial uncertainty set $\Omega_{\tilde{w}}$ as introduced in Section 3.3 (for adversarial uncertainty set only)
11:        Compute robust value target $y(s, a, r, s', \bar{w})$ by calculating the RHS of Equation 4
12:        Update action state value parameters $\theta_i$ for $i \in \{1, 2\}$ by minimizing mean squared loss to the target:
13:            $\nabla_{\theta_i} \frac{1}{|\mathcal{B}|} \sum_{(s,a,r,s') \in \mathcal{B}} (Q(s, a; \theta_i) - y(s, a, r, s', \bar{w}))^2$    for $i = 1, 2$
14:        Update policy parameters $\phi$ by policy gradient:
15:            $\nabla_{\phi} \frac{1}{|\mathcal{B}|} \sum_{(s) \in \mathcal{B}} (\min_{i=1,2} Q(s, \tilde{a}; \theta_i) - \alpha \log \pi(\tilde{a}|s; \phi))$
16:        where $\tilde{a}$ is sampled from $\pi(a|s; \phi)$ and differentiable w.r.t. $\phi$
17:        Update target value parameters:
18:            $\theta_{tar,i} \leftarrow (1 - \rho)\theta_{tar,i} + \rho\theta_i$    for $i = 1, 2$
19:     **end for**
20: **until convergence**

---

### B.4 Algorithm on Generation of Adversarial Uncertainty Set

This is the pseudo-code to generate the adversarial uncertainty set introduced in the paper.

---

**Algorithm 2** Generation of Adversarial Uncertainty Set

---

1: Input: current state $s$, current action $a$, value function $V(s; \theta)$, next state distribution $P(\cdot|s, a; \bar{w}) = \mathcal{N}(\mu(s, a; \bar{w}), \Sigma(s, a; \bar{w}))$
2: **do**
3:     Sample $\sigma \sim \mathcal{N}(0, I)$ and calculate next state $s' = \mu(s, a; \bar{w}) + \Sigma(s, a; \bar{w})\sigma$ (the reparameterization trick to sample $s' \sim P(\cdot|s, a; \bar{w})$);
4:     Forward pass to calculate the next state value $V(s'; \theta)$;
5:     Backward pass to compute the derivative $g(\bar{w}) = \nabla_w V(s'; \theta) = \frac{\partial V(s'; \theta)}{\partial s'} \frac{\partial \mu(s, a; \bar{w}) + \Sigma(s, a; \bar{w})\sigma}{\partial w}$;
6:     Normalize the derivative by $d(\bar{w}) = g(\bar{w})/[\sum_i^W g(\bar{w})_i^2]^{0.5}$;
7:     Generate the adversarial uncertainty set $\Omega_w = \{\bar{w} + \alpha\tilde{w} : \|\tilde{w}/d(\bar{w})\|_2 \leq 1\}$.
8: **done**

---

## C  Extra Experimental Setups

### C.1 Process Map of Experiments

We would like to clarify again the experimental setup in this paper. We drew a process map 5 to facilitate the understanding. During both training and testing, the agents can only acquire

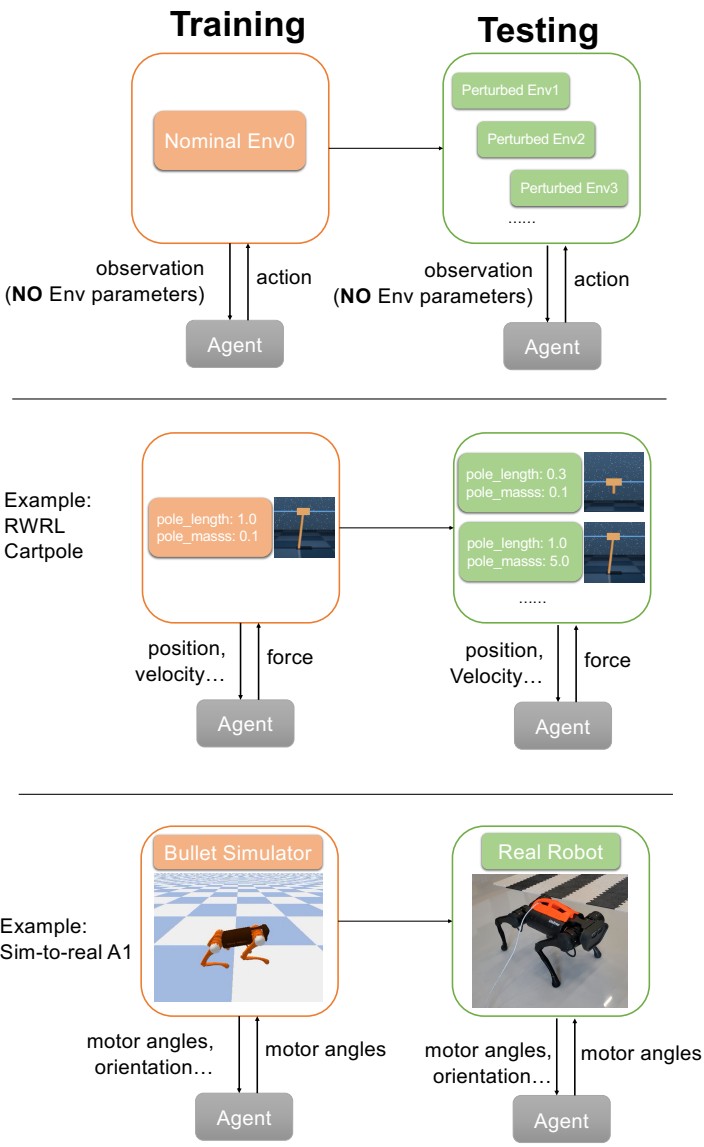

Figure 5: Process map of experiments and specific examples.

observations (e.g. position, velocity) from the environment without knowing the environmental parameters (e.g. pole length, robot mass). During training, the agents can only interact with a single set of environment parameters (nominal environment), so it's impossible to predict the pattern of the perturbation. During testing, all trained agents are fixed and tested on various perturbed environment parameters. So to speak, the agents have to learn a single policy on an unperturbed environment that can adapt to various perturbed environments, which might be far more challenging than expected. *Adv-USR* shows a stable improvement over all other baselines.

## C.2 RWRL Benchmarks

In this paper, we conduct experiments on six tasks: *cartpole_balance*, *cartpole_swingup*, *walker_stand*, *walker_walk*, *quadruped_walk*, *quadruped_run*. All tasks involve a domain and a

movement. For example, *cartpole_balance* task represents the balance movement on cartpole domain. In this paper, we consider 3 domains with 2 movements each.

The 3 domains are cartpole, walker and quadruped respectively:

- **Cartpole** has an unactuated pole based on a cart. One can apply a one-direction force to balance the pole. For *cartpole_balance* task, the pole starts near the upright while in *cartpole_swingup* task the pole starts pointing down.

- **Walker** is a planar walker to be controlled in 6 dimensions. The *walker_stand* task requires an upright torso and some minimal torso height. The *walker_walk* task encourages a forward velocity.

- **Quadruped** is a generic quadruped with a more complex state and action space than cartpole and walker. The *quadruped_walk* and *quadruped_run* tasks encourage a different level of forward speed.

For a detailed description of these tasks, please refer DM_CONTROL [51]. For each task, we follow the RWRL's setup by selecting 4 environmental variables and perturbing them during the testing phase. The perturbed variables and their value ranges can be found in Table 2. We also report the nominal values of these perturbed variables to indicate the differences between training and testing environments. Notably, all tasks are run for a maximum of 1000 steps and the max return is 1000.

Table 2: Tasks in RWRL benchmark and the perturbed variables.

| Task Name | Observation Dimension | Action Dimension | Perturbed Variables | Perturbed Range | Nominal Value |
|---|---|---|---|---|---|
| *cartpole_balance* *cartpole_swingup* | 5 | 1 | pole_length pole_mass joint_damping slider_damping | [0.3, 3.0] [0.1, 10.0] [2e-6, 2e-1] [5e-4, 3.0] | 1.0 0.1 2e-6 5e-4 |
| *walker_stand* *walker_walk* | 24 | 6 | tigh_length torso_length joint_damping contact_friction | [0.1, 0.7] [0.1, 0.7] [0.1, 10.0] [0.01, 2.0] | 0.225 0.3 0.1 0.7 |
| *quadruped_walk* *quadruped_run* | 78 | 12 | shin_length torso_density joint_damping contact_friction | [0.25, 2.0] [500, 10000] [10, 150] [0.1, 4.5] | 0.25 1000 30 1.5 |

## C.3   Practical Implementations of *Robust-AUC*

To calculate *Robust-AUC* in RWRL experiments, each agent is trained with 5 random seeds. During the testing phase, for each environmental variable $P$, we uniformly sample 20 perturbed values $v$ in the range of $[v_{min}, v_{max}]$. For each value $v$, the environment variable $P$ is first modified to value $v$ and the agent is tested for 100 episodes (20 episodes per seed). We then select the 10%-quantile [2] as the return $r$ at value $v$. By doing so we roughly have an approximated curve $C(v, r)$ and can calculate *Robust-AUC* defined previously. We also report the area between 5%-quantile and 15%-quantile as the statistical uncertainty of the reported *Robust-AUC*.

## C.4   Model Structure

The model structure for all experimental baselines is based on the Yarats and Kostrikov [52]'s implementation of Soft Actor Critic (SAC) [25] algorithm. The actor network is a 3-layer feed-forward network with 1024 hidden units and outputs the Gaussian distribution of action. The critic network adopts the double-Q structure [53] and also has 3 hidden layers with 1024 hidden units on each layer, but only outputs a real number as the action State value.

---

[2]10%-quantile as a worst-case performance evaluates the robustness of RL algorithms more reasonably than common metrics.

## C.5 Hyperparameters

To compare all algorithms fairly, we set all hyperparameters equally except the robust method and its coefficient. All algorithms are trained with Adam optimizer [26]. The full hyperparameters are shown in Table 3. For regularizer coefficients of all robust update methods, please see Table 4. Notably, the principle to choose the coefficient is to increase the value until the performance on the nominal environment drops. In the future, it can be automatically tuned by learning an unregularized value function and comparing the difference between robust value and unregularized value. All experiments are carried out on NVIDIA GeForce RTX 2080 Ti and Pytorch 1.10.1.

Table 3: Hyperparameters of Robust RL algorithms.

| HYPERPARAMETERS | VALUE | DESCRIPTION |
|---|---|---|
| BATCH SIZE | 1024 | THE NUMBER OF TRANSITIONS FOR EACH UPDATE |
| DISCOUNT FACTOR $\gamma$ | 0.99 | THE IMPORTANCE OF FUTURE REWARDS |
| REPLAY BUFFER SIZE | 1E6 | THE MAXIMUM NUMBER OF TRANSITIONS STORED IN MEMORY |
| EPISODE LENGTH | 1E3 | THE MAXIMUM TIME STEPS PER EPISODE |
| MAX TRAINING STEP | 1E6 | THE NUMBER OF TRAINING STEPS |
| RANDOM STEPS | 5000 | THE NUMBER OF RANDOMLY ACTING STEPS AT THE BEGINNING |
| ACTOR LEARNING RATE | 1E-4 | THE LEARNING RATE FOR ACTOR NETWORK |
| ACTOR UPDATE FREQUENCY | 1 | THE FREQUENCY FOR UPDATING ACTOR NETWORK |
| ACTOR LOG STD BOUNDS | [-5, 2] | THE OUTPUT BOUND OF LOG STANDARD DEVIATION |
| CRITIC LEARNING RATE | 1E-4 | THE LEARNING RATE FOR CRITIC NETWORK |
| CRITIC TARGET UPDATE FREQUENCY | 2 | THE FREQUENCY FOR UPDATING CRITIC TARGET NETWORK |
| CRITIC TARGET UPDATE COEFFICIENT | 0.005 | THE UPDATE COEFFICIENT OF CRITIC TARGET NETWORK FOR SOFT LEARNING |
| INIT TEMPERATURE | 0.1 | INITIAL TEMPERATURE OF ACTOR'S OUTPUT FOR EXPLORATION |
| TEMPERATURE LEARNING RATE | 1E-4 | THE LEARNING RATE FOR UPDATING THE POLICY ENTROPY |
| SAMPLE SIZE | 1 | THE SAMPLE SIZE TO APPROXIMATE THE ROBUST REGULARIZOR |

Table 4: Regularization coefficients of Robust RL algorithms.

| Task Name | Algorithms | | | | | |
|---|---|---|---|---|---|---|
| | None-Reg | L1-Reg | L2-Reg | L1-USR | L2-USR | Adv-USR |
| *cartpole_balance* | - | 1e-5 | 1e-4 | 5e-5 | 1e-4 | 1e-5 |
| *cartpole_swingup* | - | 1e-5 | 1e-4 | 1e-4 | 1e-4 | 1e-4 |
| *walker_stand* | - | 1e-4 | 1e-4 | 5e-5 | 1e-4 | 1e-4 |
| *walker_walk* | - | 1e-4 | 1e-4 | 1e-4 | 1e-4 | 5e-4 |
| *quadruped_walk* | - | 1e-5 | 1e-4 | 1e-4 | 1e-4 | 5e-4 |
| *quadruped_run* | - | 1e-4 | 1e-4 | 5e-5 | 1e-4 | 7e-5 |

## C.6 Extra Setups of Sim-to-real Task

We use the Unitree A1 robot [32] and the Bullet simulator [33] as the platform for sim-to-real transfer. The Unitree A1 is a quadruped robot with 12 motors (3 motors per leg). The Bullet simulator is a popular simulation tool specially designed for robotics.

It is well known that there is a non-negligible difference between simulators and real robots due to:(1) the simulator possesses a simplified dynamics model and suffers from accumulated error [31] and (2) there are significant differences between simulators and real hardware in robot's parameters, such as a quadruped example in Table 5. Therefore, training policies in simulation and applying them to real robots (sim-to-real) is a challenging task for robotics.

Specifically, we perform 2 sim-to-real tasks: standing and locomotion following previous work [34]. The detailed description of the experiment is as follows:

**Observation.** The observation contains the following features of 3 steps: motor angles (12 dim), root orientation (4 dim, roll, pitch, roll velocity, pitch velocity), and previous actions (12 dim). So the observation space is 84 dimensions.

**Action.** All 12 motors can be controlled in the position control mode, which is further converted to torque with an internal PD controller. The action space for each leg is defined as $[p - o, p + o]$. The specific values for different parts (hip, upper leg, knee) in the standing task are $p = [0.00, 1.6, -1.8], o = [0.8, 2.6, 0.8]$. The values in the locomotion task are $p = [0.05, 0.7, -1.4], o = [0.2, 0.4, 0.4]$.

**Reward.** For the standing task, the reward consists of 3 parts: $r(s, a) = 0.2 * r_{\text{HEIGHT}} + 0.6 * r_{\text{POSE}} + 0.2 * r_{\text{VEL}}$. $r_{\text{HEIGHT}} = 1 - |z - 0.2587|/0.2587$ is rewarded for approaching the standing position on the z-axis. $r_{\text{POSE}} = \exp\{-0.6 * \sum |m_{target} - m|\}$ is for correct motor positions. $r_{\text{VEL}}$ is punished for positive velocity (standing should be still in the end). For the locomotion task, the reward function is inspired by Kostrikov et al. [54]. $r(s, a) = r_v(s, a) - 0.1 * v_{\text{YAW}}^2$. $v_{\text{YAW}}$ is angular yaw velocity and $r_v(s, a) = 1$ for $v_x \in [0.5, 1.0]$, $= 0$ for $v_x \geq 2.0$ and $= 1 - |v_x - 0.5|$ otherwise, is rewarded for velocity in x-axis.

**Simulation Training.** We first train SAC agents with and without *Adv-USR* in simulations. Each step simulates 0.033 seconds so that the control frequency is 33 Hz. Agents are trained for $1e^6$ steps in the standing task and $2e^6$ steps in the locomotion task. The model structures and hyperparameters are the same as in RWRL experiments and can be referred to Appendix C.4 and C.5. The regularizer coefficients for *Adv-USR* are 1e-3 and 1e-4 for two tasks.

**Real Robot Evaluation.** After training, we directly deploy the learned policies on real robots. Since all sensors are internal on robots, there are no external sensors required. The control frequency is set as 33 Hz. We run each policy 50 episodes with 1000 steps and report the 10%-quantile of the final return and $5\% - 15\%$-quantile as the error bar in Figure 3.

Table 5: Unitree A1's Parameters in Simulation and Real Robot.

| Parameters | Simulation | Real Robot |
|---|---|---|
| Mass (kg) | 12 | [10, 14] |
| Center of Mass (cm) | 0 | [-0.2, 0.2] |
| Motor Strength ($\times$ default value) | 1.0 | [0.8, 1.2] |
| Motor Friction (Nms/rad) | 1.0 | [0.8, 1.2] |
| Sensor Latency (ms) | 0 | [0, 40] |
| Initial position (m) | (0, 0, 0.25) | ([-1, 1], [-1, 1], [0.2, 0.3]) |

# D  Extra Experimental Results

## D.1  Constant Perturbation on System Parameters

Extra experimental results for task *cartpole_balance*, *walker_walk* and *quadruped_run* can be found in Table 6. We can observe similar results as in the main paper that both *L2-USR* and *L1-USR* can outperform the default version under some certain perturbations (e.g. *L1-USR* in *cartpole_balance* for pole_mass, *L2-USR* in *walker_walk* for thigh_length), while *Adv-USR* achieves the best average rank among all perturbed scenarios, showing the best zero-shot generalization performance in continuous control tasks. Notably, *L2-Reg* in *walker_walk* and *L1-Reg* in *quadruped_run* also achieve a competitive robust performance compared with *Adv-USR*. A possible reason is that, for environments with high-dimensional state and action spaces, some of them are redundant and direct regularization on value function's parameters is effective to perform dimensionality reduction and thus learns a generalized policy.

## D.2  Computational Cost of All Algorithms

We report the average computation time (in milliseconds) for a single value update of all algorithms in Table 7. We notice that the computation of all algorithms increases as the environment's com-

Table 6: *Robust-AUC* of all algorithms and their uncertainties on RWRL benchmark.

| Task Name | Variables | Algorithms | | | | | |
|---|---|---|---|---|---|---|---|
| | | *None-Reg* | *L1-Reg* | *L2-Reg* | *L1-USR* | *L2-USR* | *Adv-USR* |
| *cartpole_balance* | pole_length | 981.45 (5.92) | **989.85 (3.74)** | 989.33 (8.32) | 798.07 (22.93) | 944.89 (23.33) | 959.66 (26.58) |
| | pole_mass | 623.88 (28.64) | 605.35 (55.60) | 607.79 (23.18) | **632.54 (14.74)** | 588.13 (38.33) | 627.00 (22.90) |
| | joint_damping | 970.83 (21.89) | 978.97 (9.95) | 982.71 (15.24) | **985.57 (10.01)** | 978.62 (17.52) | 982.43 (130.03) |
| | slider_damping | 999.44 (0.26) | 999.30 (0.43) | 999.34 (0.57) | 999.45 (0.31) | 999.49 (0.48) | **999.55 (0.32)** |
| | average rank | 4.00 | 4.00 | 3.25 | 2.75 | 4.50 | **2.50** |
| *walker_walk* | thigh_length | 315.64 (37.24) | 237.90 (25.04) | 345.12 (40.30) | 316.61 (37.86) | **350.01 (34.37)** | 318.88 (53.73) |
| | torso_length | 498.01 (54.04) | 300.39 (114.06) | 533.96 (47.73) | **550.44 (50.83)** | 543.39 (42.52) | 543.91 (54.36) |
| | joint_damping | 364.70 (50.33) | 283.19 (30.18) | **420.23 (51.84)** | 357.39 (56.04) | 356.22 (49.74) | 368.35 (64.76) |
| | contact_friction | 885.01 (27.47) | 714.94 (27.15) | **907.13 (18.94)** | 897.65 (23.49) | 900.58 (21.46) | 902.03 (24.68) |
| | average | 4.50 | 6.00 | **2.00** | 3.25 | 3.00 | 2.25 |
| *quadruped_run* | shin_length | 204.14 (91.36) | **280.11 (61.49)** | 168.95 (38.19) | 246.43 (117.07) | 214.18 (56.06) | 250.07 (79.37) |
| | torso_density | 321.24 (76.70) | **417.68 (88.55)** | 252.37 (88.41) | 319.43 (90.79) | 225.32 (80.49) | 383.14 (67.34) |
| | joint_damping | 367.05 (139.61) | 641.08 (19.12) | 687.42 (12.85) | 324.38 (14.73) | **692.02 (6.98)** | 664.25 (19.35) |
| | contact_friction | **654.43 (57.94)** | 614.21 (76.60) | 473.58 (61.72) | 632.64 (95.18) | 624.32 (124.39) | 537.19 (76.22) |
| | average rank | 3.75 | **3.00** | 5.00 | **3.00** | 3.25 | **3.00** |

Table 7: The computational cost (in milliseconds) for each value update of Robust RL algorithms.

| Task Name | Algorithms | | | | | |
|---|---|---|---|---|---|---|
| | *None-Reg* | *L1-Reg* | *L2-Reg* | *L1-USR* | *L2-USR* | *Adv-USR* |
| *cartpole_balance* *cartpole_swingup* | 14.72 ± 1.57 | 16.48 ± 1.68 | 17.05 ± 1.63 | 17.62 ± 1.49 | 22.48 ± 3.21 | 22.48 ± 3.21 |
| *walker_stand* *walker_walk* | 15.71 ± 1.23 | 18.89 ± 1.58 | 17.52 ± 1.92 | 18.06 ± 1.80 | 18.39 ± 1.90 | 23.16 ± 1.72 |
| *quadruped_walk* *quadruped_run* | 15.93 ± 1.68 | 19.47 ± 1.47 | 19.56 ± 1.67 | 20.79 ± 4.20 | 19.13 ± 1.98 | 25.23 ± 2.14 |

plexity grows, and *L1-Reg*, *L1-Reg*, *Adv-USR*'s complexities are acceptable compared with other baselines (× 1 ∼ 1.25 time cost). The computation only becomes a problem when applying *USR* methods to dynamics with millions of parameters (common in model-based RL [38]). To tackle this issue, we can identify important parameters to reduce computation costs, as stated in Section 6.

Theoretically, the additional computational cost largely depends on the norm term $\|\nabla_w P(s'|s, a; \bar{w}) V^\pi(s')\|_2$ in Equation 5, time complexity is $O(W)$ ($W$ is the number of parameters).

## D.3 Constant Perturbation on Multiple System Parameters

In real-world scenarios, there would be uncertainties in all system parameters. We provide the following additional experimental results to show the robustness when all parameters are perturbed simultaneously. The specific environmental setup is that all 4 parameters are perturbed simultaneously during testing. The perturbation intensity grows from 0 to 1. 0 resembles training environments without perturbations and 1 represents the allowed maximum perturbed values in Table 2. We adopt the same metric *Robust-AUC* and report it in the following table. All methods become less robust due to the increasing difficulty of perturbations, but *Adv-USR* still outperforms others.

Table 8: *Robust-AUC* of all algorithms and their uncertainties on RWRL benchmark.

| Task Name | Algorithms | | | | | |
|---|---|---|---|---|---|---|
| | *None-Reg* | *L1-Reg* | *L2-Reg* | *L1-USR* | *L2-USR* | *Adv-USR* |
| *cartpole_swingup* | 867.42 (0.27) | 856.87 (0.52) | 866.99 (0.23) | 867.61 (0.44) | 867.45 (0.26) | **881.36 (0.21)** |
| *walker_stand* | 254.04 (32.91) | 235.36 (25.29) | 254.64 (35.01) | 262.57 (25.02) | 263.35 (24.85) | **266.97 (7.75)** |
| *quadruped_walk* | 522.98 (34.93) | 524.24 (85.03) | 525.51 (24.58) | 525.14 (85.68) | 506.17 (54.92) | **534.61 (18.25)** |

### D.4 Noisy Perturbation on System Parameters

One may also be interested in the noisy perturbation setup where system parameters keep changing at every time step. This setup extends the Robust RL framework where the perturbation is fixed throughout the whole episode. The specific experimental setup noisy perturbation is as follows: the environmental parameter starts from the nominal value and follows a zero-mean Gaussian random walk at each time step. The nominal value and the standard deviation of the Gaussian random walk are recorded in Table 9. The experimental result on *quadruped_walk* is shown in Figure 6. In this experiment, *L1-Reg* achieves the best robustness, while our method *Adv-USR* achieves Top-2 performance in 3 out of 4 perturbations. While *L1-Reg* performs less effectively in the case of fixed perturbation, it implies that different regularizers do have different impacts on these two types of perturbations. For noisy perturbation, environmental parameters walk randomly around the nominal value and reach the extreme value less often, which requires a less conservative robust RL algorithm. Our algorithm *Adv-USR*, originally designed for fixed perturbation problem, achieves good but not the best performance, which leads to an interesting future research direction on the trade-off between robustness and conservativeness.

Table 9: The perturbed variables for the noise perturbation experiment.

| Task Name | Perturbed Variables | Start Value | Step Standard Deviation | Value Range |
|---|---|---|---|---|
| *quadruped_walk* | shin_length | 0.25 | 0.1 | [0.25, 2.0] |
| | torso_density | 1000 | 500 | [500, 10000] |
| | joint_damping | 30 | 10 | [10, 150] |
| | contact_friction | 1.5 | 0.5 | [0.1, 4.5] |

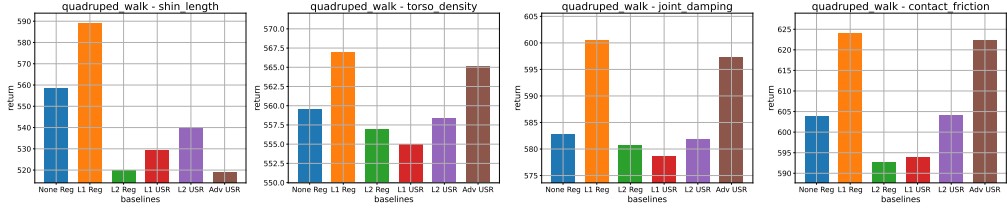

Figure 6: The parameter-return bar graph of all algorithms. All bars represent 10%-quantile value of episodic return under noisy environmental parameters.

### D.5 Training Performances

We further analyze the effects of *Adv-USR* during the training process. Since all algorithms are trained on the same nominal environment, they all learn to perform well on the nominal environment with a similar episode reward curve as in Figure 7a. However, their target values considering the robustness of learned policies are quite different (Figure 7b). *Adv-USR* has the lowest target value, indicating that the adversarial uncertainty set actually considers the pessimistic objective which encourages to learn more robust policy to resist this uncertainty. This verifies the correctness of our theoretical claims.

## E   Comparison with Domain Randomization

The rethink on domain randomization (DR) directly motivates this paper: domain randomization utilizes a variety of environments and trains an average model across them. Could we develop a more efficient way? Here we discuss the drawbacks of DR and why our method could be a possible solution.

- **Requirements on expert knowledge:** DR needs to randomize multiple environments with various environmental parameters. However, which parameters and their range to random-

ize all require heavy expert knowledge. We have experience when applying DR on the sim-to-real locomotion task. If we set the PD controller's gain in a large range, RL fails to learn even in simulation. If this range is small, the learned policy can't transfer to real robots. In comparison, our method reduces this effort by only training a robust policy on a single nominal environment with a virtual adversarial uncertainty set.

- **Feasibility on certain setups:** In some cases, we don't have access to create multiple simulated environments. This could happen in commercial autonomous driving software without exposing the low-level dynamics system, or the current powerful ChatGPT-styled cloud-based language model (if viewing the chatbot as a simulator), or even real-to-real transfer (from one real robot to another slightly different real robot). It's not possible to create different environments for DR but still possible to learn an adversarial uncertainty set and train a robust policy based on that.

- **Training convergence:** In general, DR has a slower and more unstable training process since randomized multiple environments can be quite different and learning a single policy on them can be hard.

We compare the popular domain randomization (DR) techniques on sim-to-real tasks (A1 quadruped standing and locomotion). The experimental setup is as follows. During **training in simulation**, for *None-Reg* and *Adv-USR*, agents are trained on the nominal environment with standard robots' parameters ("Other" column in Table 10). For *DR*, agents are trained on randomized initialized parameters within a certain range ("DR" column in Table 10). During **testing on real robots**, all policies are fixed and evaluated on the same robots with 50 episodes. We present the training curves in Figure 8 and testing performance in Figure 9.

DR has a slower convergence rate during training since randomized environments can be quite different and learning a single policy on them can be hard. *Adv-USR* can already replace DR on simple sim-to-real tasks (standing). For more complex tasks (locomotion), DR still performs better since the performance is more affected by the model mismatch. But we believe our method is still valuable to provide an alternative concise choice to DR. Furthermore, as we have mentioned in the limitation section, our method is not perpendicular to the DR. Combing adversarial uncertainty sets could potentially reduce the range of randomization of DR while reaching the same robust performance.

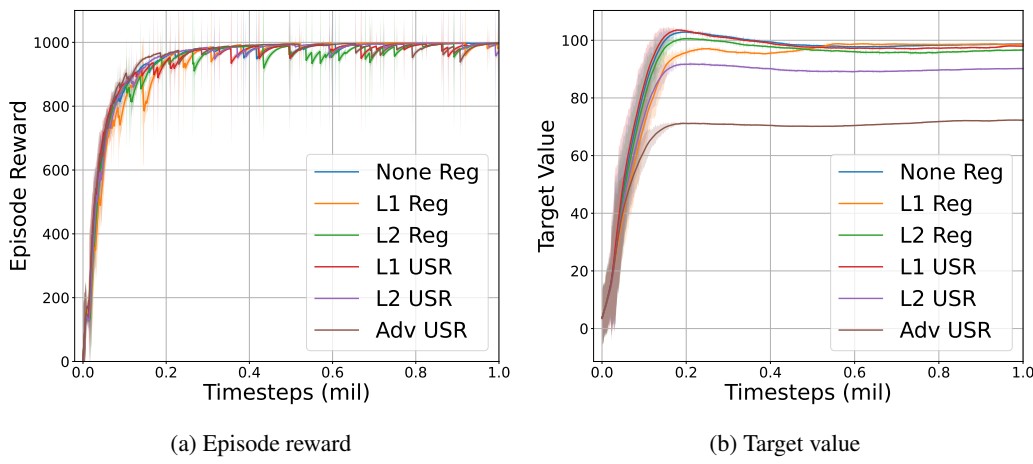

(a) Episode reward           (b) Target value

Figure 7: Training performances of all algorithms on *quadruped_walk* task
.

Table 10: Unitree A1's Parameters in Simulation for different algorithms.

| Parameters | Other | DR |
|---|---|---|
| Mass (kg) | 12 | [10, 14] |
| Center of Mass (cm) | 0 | [-0.2, 0.2] |
| Motor Strength ($\times$ default value) | 1.0 | [0.8, 1.2] |
| Motor Friction (Nms/rad) | 1.0 | [0.8, 1.2] |
| Sensor Latency (ms) | 0 | [0, 40] |
| Initial position (m) | (0, 0, 0.25) | ([-1, 1], [-1, 1], [0.2, 0.3]) |

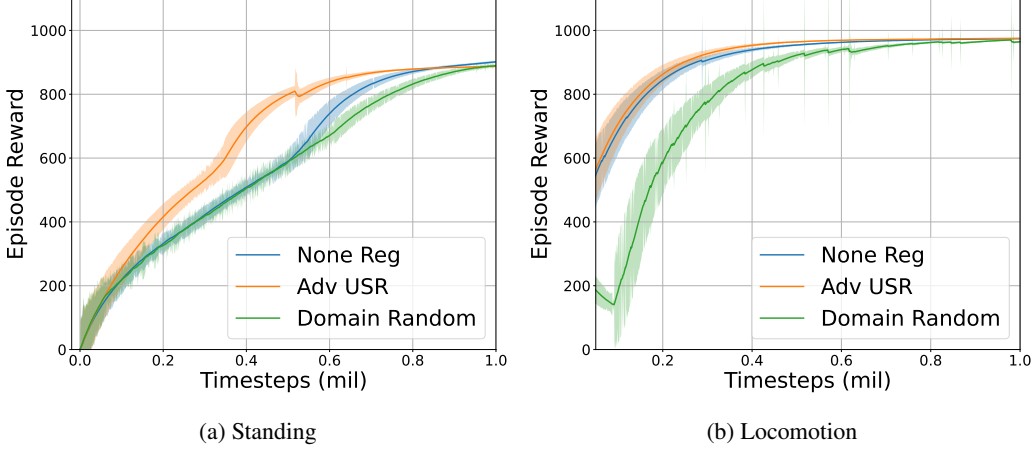

(a) Standing

(b) Locomotion

Figure 8: Episode returns during training (simulation).

## F Comparison with State Perturbation

State perturbation describes the uncertainties in the output space of the dynamics model, which is a special case of transition perturbation. We illustrate how to transform the State perturbation into transition perturbation in the following case.

Considering a $L_2$ uncertainty set on the output of the dynamics model, $\Omega_{s_p} = \{s_p | s' \sim P(\cdot|s, a; \bar{w}), \|s' - s_p\| \leq 1\}$. We can rewrite the next State distribution $s' \sim P(\cdot|s, a; \bar{w})$ as $s' = f(s'|s, a; \bar{w}) + \eta$, where $\eta$ is a random noise $\eta \sim \mathcal{N}(0, 1)$. Then the perturbed State can be written as $s_p = f(s'|s, a; \bar{w}) + \eta + \beta, \|\beta\| \leq 1$. Viewing $\beta$ as one additional parameter in the dynamics model, the uncertainty set on $\beta$ is actually $\Omega_\beta = \{\|\beta\| \leq 1\}$. Based on this uncertainty set

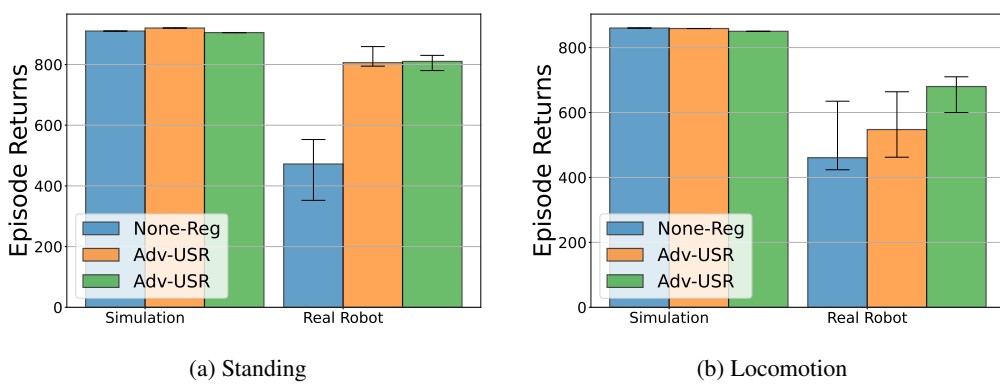

(a) Standing

(b) Locomotion

Figure 9: Episode returns during testing (simulation and real robot).

on $\beta$, one can further design corresponding regularizers on value function to increase the robustness as discussed in the paper.

If the uncertainty set on state space is unknown, denoted as $\Omega_\beta$, it is still feasible to include $\beta$ as an additional parameter in the dynamics model. As a result, *Adv-USR* could still be used to handle this unknown uncertainty set on $\beta$.

