# OpenReview forum: "Robust Reinforcement Learning in Continuous Control Tasks with Uncertainty Set Regularization"
_robot-learning.org/CoRL/2023/Conference — CoRL 2023 Poster_

### Official Review · Reviewer_5r3v · 2023-07-07

**Confidence:** 3
**Originality:** Good
**Technical Quality:** Good
**Clarity Of Presentation:** Good
**Impact:** 2

**Recommendation:**

Weak Accept: I recommend accepting the paper, but will not argue for my recommendation if the majority of other reviewers have a different opinion.

**Review:**

Strengths:
- The paper seems interesting and it seems to have some novel contributions, e.g. adversarial uncertainty set, incorporated USR into the policy interation framework.
- The method seems to be pretty effective compared to baselines in the evaluations.
- On the whole, the paper and appendix are well written.

Weaknesses:
- The method relies on learning a local transition model but this is likely nontrivial to learn in many scenarios, such as vision-based learning systems. If there is not sufficient coverage of data for this model, it will likely be quite inaccurate.
- It would be good to see some additional empirical analyses and ablations, especially on how the approximated target value using USR-Adv compares to the baseline methods and the sensitivity to hyperparameters and amount/coverage of data.
- It is a bit confusing what the authors want to focus on as the key takeaway and the main contribution. The main (first) contribution emphasized is extending the robustness-regularization duality to continuous control tasks, but this is not a large contribution or a difficult derivation. I think the authors should focus more on a clearer takeaway.
- To clarify, are the environment parameters required of all data during training? I think if so, this is a large assumption that should be emphasized.

**Quality Of The Limitations Section:**

Additional details required

**Questions For Rebuttal:**

See weaknesses above.

**Robotics Focus:**

Sufficient demonstration on hardware

**Summary Of Paper:**

This paper studies the problem setting of robust reinforcement learning, where the goal is to learn a robust policy that performs well in the worst-case environment dynamics. It proposes Uncertainty Set Regularized Reinforcement Learning, which extends the robustness-regularization duality to a continuous state space and incorporates it into a policy iteration framework. The authors propose generating an adversarial uncertainty set aimed at performing better against perturbations. The paper includes results on the Real-world RL benchmark along with sim-to-real results on an A1 robot and proposes a new evlaution metric, Robust-AUC, for evaluating policy robustness.

**Summary Of Recommendation:**

I think this paper has some promising ideas and results but is not quite at publication level and would benefit from additional analyses and clarifications.

---

> ### Author Response · Authors · 2023-08-15
> **Authors' Response**
>
> Dear Reviewer,
>
> As the discussion period is almost to the end, could you please check whether our response has addressed your concerns? If not, we are pleased to give further explanation.

---

### Official Review · Reviewer_2mmx · 2023-07-17

**Confidence:** 3
**Originality:** Fair
**Technical Quality:** Fair
**Clarity Of Presentation:** Good
**Impact:** 2

**Recommendation:**

Weak Accept: I recommend accepting the paper, but will not argue for my recommendation if the majority of other reviewers have a different opinion.

**Review:**

The main concerns are:
1. the paper structure should be updated. It is suggested to summarize the algorithm process in paper rather than appendix. Also it is recommaned to revise the paper to proper introduce the related works in paper.
2. the experimental results should be hugely improved to let reader fully understand the superiority of USR, especially adv-USR. Besides the Robust-AUC, what is the positive impact of adv-USR to the agent's behavior and the corresponding learning curves?
3. The real experiment was less introduced, please detail the result and demonstrate the advantages of Adv-USR clearly.

**Quality Of The Limitations Section:**

Additional details required

**Questions For Rebuttal:**

Please see my comments above.

**Robotics Focus:**

Highly relevant to robotics but no hardware experiments

**Summary Of Paper:**

This paper tackle the issue of robustness in robot reinforcement learning with various environmental settings by introducing the uncertainty set regularizer. The proposed method successfully implemented USR to SAC and demonstrated its superiority in both Mujoco simulation and simple real robot settings.

**Summary Of Recommendation:**

I recommanded a weak reject based on my comment above.

---

> ### Author Response · Authors · 2023-08-15
> **Authors' Response**
>
> Dear Reviewer,
>
> As the discussion period is almost to the end, could you please check whether our response has addressed your concerns? If not, we are pleased to give further explanation.

---

### Official Review · Reviewer_8DAc · 2023-07-19

**Confidence:** 3
**Originality:** Good
**Technical Quality:** Very Good
**Clarity Of Presentation:** Very Good
**Impact:** 3

**Recommendation:**

Weak Accept: I recommend accepting the paper, but will not argue for my recommendation if the majority of other reviewers have a different opinion.

**Review:**

This paper is well-written and clear in its presentation, experiments, and ideas mostly. Many of the ideas are derived from the duality in the tabular MDP setting, so not completely original, but still has some novel ideas such as automatic tuning of uncertainty sets. The problem being tackled is quite significant as addressing sim-to-real is a big challenge for RL, and the authors do a good job of motivating this.

<Strengths>

1. Overall, the structure and the organization of the paper is quite easy to follow, and relevant concepts are adequately introduced to the reader.

2. The continuous-space extension of an existing result in discrete state-action MDP is a valuable contribution.

3. The authors do extensive experimentation of the benchmark to highlight  the efficacy of the approach.

<Weaknesses>

1.  The authors should make some convincing argument of what robustness offers over domain randomization. The authors remarked about the speed which is convincing, but there are no empirical comparisons against DR that shows that this approach is more computationally tractable or results in more “robust” performance.

2. The authors have a strong theoretic claim on the Bellman iteration but the quantities of the approximation rely on function approximation, which can destroy the claims of the theory in practice.

3. The uncertainty structure that the authors can handle is still somewhat limited to relatively structured uncertainty.

<Minor Suggestions>

1. In the abstract, “Prior work claimed that adding regularization to the value function is equivalent to learning a robust policy with uncertain transitions.” - I think it’d be better to say which kind of regularization explicitly because regularization is too broad of a term. Even adding noise to transition dynamics by itself is also a form of regularization since it makes the value function smooth. .

2. It would be good to explicitly say how (2) differs from typical Bellman by incorporating Psa. (e.g. note that different from usual Bellman, there is this term that accounts for minimax).

3. When describing uncertainty perturbation by \bar{P}_{sa} + \tilde{P}_{sa},  What is a norm on \tilde{P}_sa? P_{sa} is a probability distribution, do authors mean some kind of function norm on a distribution? The notation describing Psa is very confusing in general, it’s unclear what the authors mean by cross in line 68. I understood the high-level idea that it’s a state-action independent rectangular bound.

4. Where is algorithm 2?

5. Can we be more precise about which kind of uncertainty sets the algorithm can handle? The authors utilize an ellipsodial representation but it would be nice to be fully general.

**Quality Of The Limitations Section:**

Limitations are addressed clearly

**Questions For Rebuttal:**

1. How would the authors compare their algorithm against domain randomization? Besides being more faster (requiring less parallelization), would it result in more robustness at the cost of performance? Why would folks choose this algorithm over DR?

2. I did not fully understand the minimax duality mathematically, but in the usual context of optimization, there can be a duality gap between the function and its dual function (e.g. convex conjugate) in the absence of structure. What prevents the authors from solving a regularized problem and ending up with a totally different solution from the primal minimax problem?

3. As the authors mentioned, usually a difficult point in robust control lies in its conservativeness; it seems the adversarial uncertainty set procedure gives guidance for the shape of the ellipsoids, but one still needs to choose an alpha. Is this one experimentally done?

4. I am a bit concerned about the model learning part, as a lot of results for RL indicate that model learning is fundamentally related to the distribution of data it was trained on. What prevents the bellman iteration from going unstable because of errors in estimating the model and its gradients, as well as being subject to distribution shift?

**Robotics Focus:**

Sufficient demonstration on hardware

**Summary Of Paper:**

The authors propose an algorithm that solves minimax RL where the inner maximization happens over perturbative transitions of the environment. The authors rely on previous works that convert the primal minimax form into a tractable dual form and extend the work to continuous domains. Furthermore, the authors make an ellipsoidal approximation to the uncertainty set by querying the sensitivity of the value function with respect to different directions in the uncertainty set. The authors compare their approach to non-robust RL and prove robust performance of the method.

**Summary Of Recommendation:**

This paper is strong in presentation and technical details, which is why I recommend weak accept. However, I did not choose strong accept but the method itself did not seem sufficiently motivated for me other than being a possible application of a mathematical regularization technique. I believe the paper would be stronger if the authors connected their method back to the original motivation of asking: if the goal is to solve robust RL, what is the best solution we can find? In an age where the vast majority is domain randomization for sim-to-real transfer, I believe it's important for the authors to show very convincingly offers over DR, and the comparison felt a bit lacking.

EDIT: I have read the author's response and found the arguments convincing, and I recommend that the paper be accepted.

---

### Official Review · Reviewer_Fg3n · 2023-07-19

**Confidence:** 3
**Originality:** Good
**Technical Quality:** Very Good
**Clarity Of Presentation:** Very Good
**Impact:** 4

**Recommendation:**

Weak Accept: I recommend accepting the paper, but will not argue for my recommendation if the majority of other reviewers have a different opinion.

**Review:**

At the beginning of the paper authors clearly motivates the importance of robust reinforcement learning methods and introduce robust RL formalism. After that, they describe additional term added to the Bellman equation which forces the robustness of the learned policy. That term requires a transition model P(s'|s,a;w) which is modeled as Gaussian distribution. The authors compare the proposed method with other regularization methods on simulated benchmarks using AUC-based metric. The authors also demonstrate that the proposed method can be used to bridge the sim-to-real gap in Unitree A1 for simple tasks without domain randomization during training.

**Quality Of The Limitations Section:**

Limitations are addressed clearly

**Questions For Rebuttal:**

The paper presents an interesting method to train robust reinforcement learning agents. I think it would be interesting to compare the performance of the proposed method and the common domain randomization technique in the benchmarks.

Also, there is a typo in the Abstract (Algorithm 1): "Construct adversarial adversarial uncertainty set"

**Robotics Focus:**

Sufficient demonstration on hardware

**Summary Of Paper:**

The paper tackles the task of robust reinforcement learning in continuous control environments. The authors generalize a robust reinforcement learning approach to continuous control and propose a new regularizer named Uncertainty Set Regularizer. For unknown uncertainty sets authors propose an adversarial approach based on the value function. The authors evaluated the proposed approach for the Soft Actor Critic method and presented evaluation results for simulated benchmarks and for the Unitree A1 robot.

**Summary Of Recommendation:**

I recommend accepting the paper after addressing questions.

---

### Author Response · Authors · 2023-08-11
**Letter to AC**

Dear Area Chair,

First, we are thankful for your work. We reach out to you is due to the misunderstanding of two reviewers to our paper. We summarize the points as follows:

* Reviewer 2mmx is concerned with the paper's structure, which we believe it's a personal preference instead of the weakness of the paper. The reviewer asks for "agent's behavior" of the proposed method, which has already been demonstrated by the videos in the supplementary material. Besides, the comments on the real-world experiments are abstract with missing details ("The real experiment was less introduced"), which confuses us in the rebuttal.

* Reviewer 5r3v makes factual errors in the experimental setup ("Are the environment parameters required of all data
during training?" ), which could lead to an underestimate of our contribution.

We wish you could consider this case. We appreciate your help in advance.

Best wishes, the authors

---

> ### Comment · Area_Chair_yzLW · 2023-08-13
> **Re: Letter to AC**
>
> Thanks, I'll keep these in mind.

---

### Decision · Program_Chairs · 2023-08-30

**Decision:**

Accept (Poster)

**Comment:**

The reviewers agree that this paper makes a worthy contribution to CoRL -- it introduces a method for robust RL that can handle continuous domains, and validates the method in both simulation and on a real unitree A1 robot. I encourage the authors to incorporate the reviewer feedback in the final version. This includes incorporating the new experiments into the paper, clarifying points of confusion, and including qualitative results from the real robot experiments.